# Childhood brain morphometry in children with persistent stunting and catch-up growth

**Beena Koshy**[1]*, **Vedha Viyas Thilagarajan**[1,2], **Samuel Berkins**[3], **Arpan Banerjee**[3], **Manikandan Srinivasan**[4], **Roshan S. Livingstone**[5], **Venkata Raghava Mohan**[2], **Rebecca Scharf**[6], **Anitha Jasper**[5], **Gagandeep Kang**[2,4]

1 Developmental Paediatrics, Christian Medical College, Vellore, Tamil Nadu, India, 2 Department of Community Health, Christian Medical College, Vellore, Tamil Nadu, India, 3 National Brain Research Centre, Gurgaon, Haryana, India, 4 Wellcome Trust Research Laboratory, Christian Medical College, Vellore, Tamil Nadu, India, 5 Department of Radiodiagnosis, Christian Medical College, Vellore, Tamil Nadu, India, 6 Centre for Global Health, University of Virginia Children's Hospital, Virginia, Virginia, United States of America

* beenakurien@cmcvellore.ac.in

## Abstract

### Background

Early childhood stunting affects around 150 million young children worldwide and leads to suboptimal human potential in later life. However, there is limited data on the effects of early childhood stunting and catch-up growth on brain morphometry.

### Methods

We evaluated childhood brain volumes at nine years of age in a community-based birth-cohort follow-up study in Vellore, south India among four groups based on anthropometric assessments at two, five, and nine years namely 'Never Stunted' (NS), 'Stunted at two years and caught up by five years' (S2N5), 'Stunted at two and five years and caught up by nine years' (S2N9), and 'Always Stunted' (AS). T1-weighted magnetic resonance imaging (MRI) images were acquired using a 3T MRI scanner, and brain volumes were quantified using FreeSurfer software. Analysis of Variance (ANOVA) was used to determine the differences in brain volumetry between the stunting groups, with age and sex as covariates. The effect size ANOVA models was evaluated using Eta squared.

### Findings

Amongst 251 children from the initial cohort, 178 children with a mean age of 9.54 underwent neuroimaging and considered for further analysis. The total brain volume, subcortical volume, bilateral cerebellar white matter, and posterior corpus callosum showed a declining trend from NS to AS. Regional cortical brain analysis showed significant lower bilateral lateral occipital volumes, right pallidum, bilateral caudate, and right thalamus volumes between NS and AS.

**Data Availability Statement:** Anonymised dataset is uploaded in the Harvard Dataverse - https://doi.org/10.7910/DVN/KJGMAY.

**Funding:** 1. Bill and Melinda Gates Foundation The
Etiology, Risk Factors and Interactions of Enteric
Infections and Malnutrition and the Consequence
for Child Health and Development Project (MAL-
ED) is carried out as a collaborative project
supported by the Bill and Melinda Gates
Foundation, the Foundation for the NIH and the
National Institutes of Health/Fogarty International
Center (Grant number – OPP 47075) 2. DBT-
Wellcome Trust India Alliance The 9-year follow-up
of the Mal-ed India cohort was supported by an
Intermediate clinical and public health research
fellowship awarded by the DBT/Wellcome Trust
India Alliance to Dr. BK. (Fellowship grant number
IA/CPHI/19/1/504611) The funders had no role in
the study design, data collection and analysis,
decision to publish or preparation of manuscript.

**Competing interests:** The authors have declared
that no competing interests exist.

## Interpretation

To the best of our knowledge, this first neuroimaging analysis to investigate the effects of persistent childhood stunting and catch-up growth on brain volumetry indicates impairment at different brain levels involving total brain and subcortical volumes, networking/connecting centres (thalamus, basal ganglia, callosum, cerebellum) and visual processing area of lateral occipital cortex.

## Introduction

Early childhood stunting affecting millions of children worldwide, is a significant risk to sustainable development goals (SDGs) planned to achieve by 2030 [1, 2]. Childhood stunting, defined as height/length standard deviation scores less than -2 as per the WHO growth charts for age and sex, affects more than 150 million children globally [1, 3, 4]. Stunting is found to be negatively associated with childhood development, cognition, school performance and later economic potential, and can result in a vicious cycle of intergenerational propagation of poverty [5–7]. Considering the sizable and persistent impact of early childhood stunting, possible mechanisms of influence are explored not just to develop effective interventions to improve childhood development/cognition but also to understand the interplay of complex childhood environments and factors.

Early childhood, including the first 1000 days of life (up to two years of age), is crucial for brain development. The development of the central nervous system starts as early as the third week of gestation with the development of the notochordal process, with further neuronal development, migration, differentiation, synaptogenesis, synaptic pruning and myelination happening in the first 1000 days of an individual's life [8, 9]. The significant growth and development of the brain in early childhood make it vulnerable to ongoing risks including malnutrition and concurrent stunting, a possible mechanism to explain the life-long effects of early childhood stunting [4, 7, 9]. However, there is a dearth of direct evaluation of brain morphology in association with early childhood stunting, which can aid in a more accurate interpretation of the neurological impact of early childhood malnutrition [10, 11]. Most studies evaluating concurrent and long-term effects of childhood stunting have evaluated indirect brain functions such as child development, child cognition, problem solving skills, behaviour and school performance [10, 11]. In addition to protein and energy deficiency, childhood stunting can be also associated with macro and micro-nutrient deficiencies with corresponding direct effects on the brain, the studies of which are also limited [9].

The high-end cost of neuroimaging measures such as magnetic resonance imaging (MRI), complex hospital-based installation practices and technical sophistication in their image interpretation had been a barrier in direct brain analysis in low-and-middle-income country (LMIC) community settings where malnutrition and childhood stunting are more prevalent [10, 11]. Direct brain measurements such as brain volumetry, spectroscopy, tractography analysis and functional assessments using MRI can help in understanding nuances of childhood malnutrition and stunting including possible causative pathways and compensations, but there is only limited literature available due to the afore-mentioned limitations.

Given this background, the objective of the current study was to evaluate brain volumetry in mid-childhood (ages 9–11 years) using a LMIC birth cohort follow-up study to understand neurological outcomes of early childhood stunting and catch-up growth during the childhood period of brain maturation. We hypothesised that brain volumes would be lower in children

with early childhood stunting compared to those not stunted at two years of age with possible improvements for those with later catch-up.

## Materials and methods

### Settings and participants

The town of Vellore, with a population of 601,000 residents situated in south India at 12.9˚ N, 79.1˚ E, has been chosen to serve as the research study site. The birth cohort in Vellore that was initially enrolled for the 'Malnutrition and Enteric Diseases' (MAL-ED) Network made up the current study population [12]. Within the population of 12000 individuals covered between 2010 and 2012, 301 eligible mothers were identified. Exclusion criteria encompassed existing plans for the family to migrate away from the study site during the specified study period, instances of multiple pregnancies, another child already enrolled in the study, medical co-morbidities in the index child, and unavailability of mothers to provide necessary informed consent. Following exclusions, the enrolled children were subject to follow-up at various time intervals, including at ages two, five, and nine years [13–15]. At each level, enrolment of the child was conditional upon written informed consent from parents, with additional child assent required at the age of nine. The initial birth cohort recruitment and subsequent follow-ups were approved by the Institutional Review Board and Ethics Committee of Christian Medical College Vellore, India.

### Measures

Up until the age of two, children's lengths were measured using an infantometer. After two years, the height was measured using a stadiometer. Every measurement was made to the nearest centimetre. Additional details regarding measurements, standardisation and follow-up can be found in earlier publications [6, 13, 14]. Multicentre Growth Reference Study (MGRS) criteria were used for measures taken at or before the age of five, and WHO AnthroPlus software was used for data taken at age nine to calculate the z-scores for height for age to determine stunting. Weight was measured using a computerised scale with an accuracy rating of ten grams.

### MRI image acquisition and processing

MRI scans were performed for the study participants at nine years of age using a Siemens Skyra 3T MRI scanner. High resolution 3D volumetric T1w images were acquired using the magnetization-prepared rapid gradient-echo (MP-RAGE) sequence with (Repetition time (TR) = 2350 ms, Echo time (TE) = 1.74ms; Inversion time (TI) = 1100ms and non-selective excitation at 7˚, Field of view (FOV) = 256 mm, slice thickness = 1 mm) following the tested protocols recommended for use with (https://www.nmr.mgh.harvard.edu/~andre/FreeSurfer_recommended_morphometry_protocols.pdf) Freesurfer™ version six. Freesurfer has been chosen for this study due to its widespread acceptance as a standard for brain volumetric studies [15] and its validation for use in children aged 4 to 11 years [16].

All images were visually inspected for quality, artifacts and clinical abnormalities using the Freeview tool before segmentation. The quality control of visual scans was also assessed using Qoala-T, a supervised-learning tool, that has been developed based on 784 (T1) scans of subjects aged from 8 to 25 years [17], and parcellated into 68 regions based on Desikan-Killiany atlas using FreeSurfer™ version six software. The selection of the Desikan-Killiany atlas for this study was based on its consistent representation of atrophy levels and accurate depiction of the

macroscopic anatomy of gyri and sulci. This made it a preferred choice over other atlases for examining both healthy children and those with stunted growth [18, 19].

The project psychologist was responsible for guiding the participants through the MRI acquisition process and for scheduling rescans, if needed. Parents had the option to opt out of receiving additional information sheets and consent details on the MRI brain analysis, which were given to all parents and children. A few days prior to children being brought in for the MRI scan, the families received thorough information and preparation after consent/assent from parents and children. Children received an MRI brain scan without general anaesthesia, and nervous children were given a repeat appointment for another MRI scan. If, after proper preparation the second time, the children were nervous, they were given a single dosage of Triclofos syrup, up to a maximum of 15 ml (1500 mg), at a rate of 0·5 ml (50 mg)/kg, and the MRI was completed when they were asleep.

## Statistical analyses

Children were divided into four categories based on their stunting status at two, five, and nine years of age: Group NS: children who were never stunted, Group S2N5: children who were stunted at two years with a catch-up at five years, and Group S5N9: children stunted at two and five years, with a catch-up at nine years, and Group AS: children stunted at two, five, and nine years [20]. Details of baseline demographic information and clinical characteristics were summarised. All brain parcellation volumes were presented as mean volumes (mm$^3$) with 95% confidence interval. Model building looked at potential confounding variables that were pre-identified based on clinical and demographic measurements of the child's age and sex. Analysis of variance (ANOVA) was used to determine the statistical significance of test results between the groups. With age and sex acting as a covariate, a multivariate ANOVA was employed to examine the differences in the overall brain regions of the various groups according the stunting status. We also examined the effect of stunting on different brain regions and the CPM score by using the Eta squared to estimate effect size in ANOVA models. Cohen's D statistics was used to calculate effect size of brain volume differences between the groups. The statistical analysis was performed with Stata version 17 software. Adjustments for multiple comparison were carried out using the Benjamin-Hochberg procedure.

## Results

Between 2010 and 2012, a survey of the study site recruited willing pregnant women enlisting 251 new-borns for the MAL-ED India cohort. Fifty children were excluded from enrolment for various reasons, including pre-existing plans for migration (five), multiple pregnancies (one), existing enrolment of another child in the study (eight), medical co-morbidities in children (seven), unavailability of the mother (nine), a combination of two or more of the aforementioned reasons (10), and maternal refusal to participate (10). Subsequently, 228 children were assessed at two years of age, 212 children at five years of age, and 205 at nine years of age [13, 16]. Some of the children and their families moved out of the study area during the follow-up research, therefore they were not included in the analysis. The nine-year recruitment was planned between February 2019 and February 2021 but was paused for more than six months in 2020, due to the Covid-19 pandemic. Anthropometric, cognitive, and behavioural assessments were completed by April 2021 and MRI brain analysis by December 2021.

Of the 205 children remaining, 14 of them declined to have an MRI, and six of them refused to cooperate when the MRI was being acquired. As a result, 185 kids out of 205 underwent MRI scans (Fig 1). Among the 185 kids, 12 of them were given Triclofos syrup for sedation. Three more children were eliminated following MRI acquisition because of incomplete

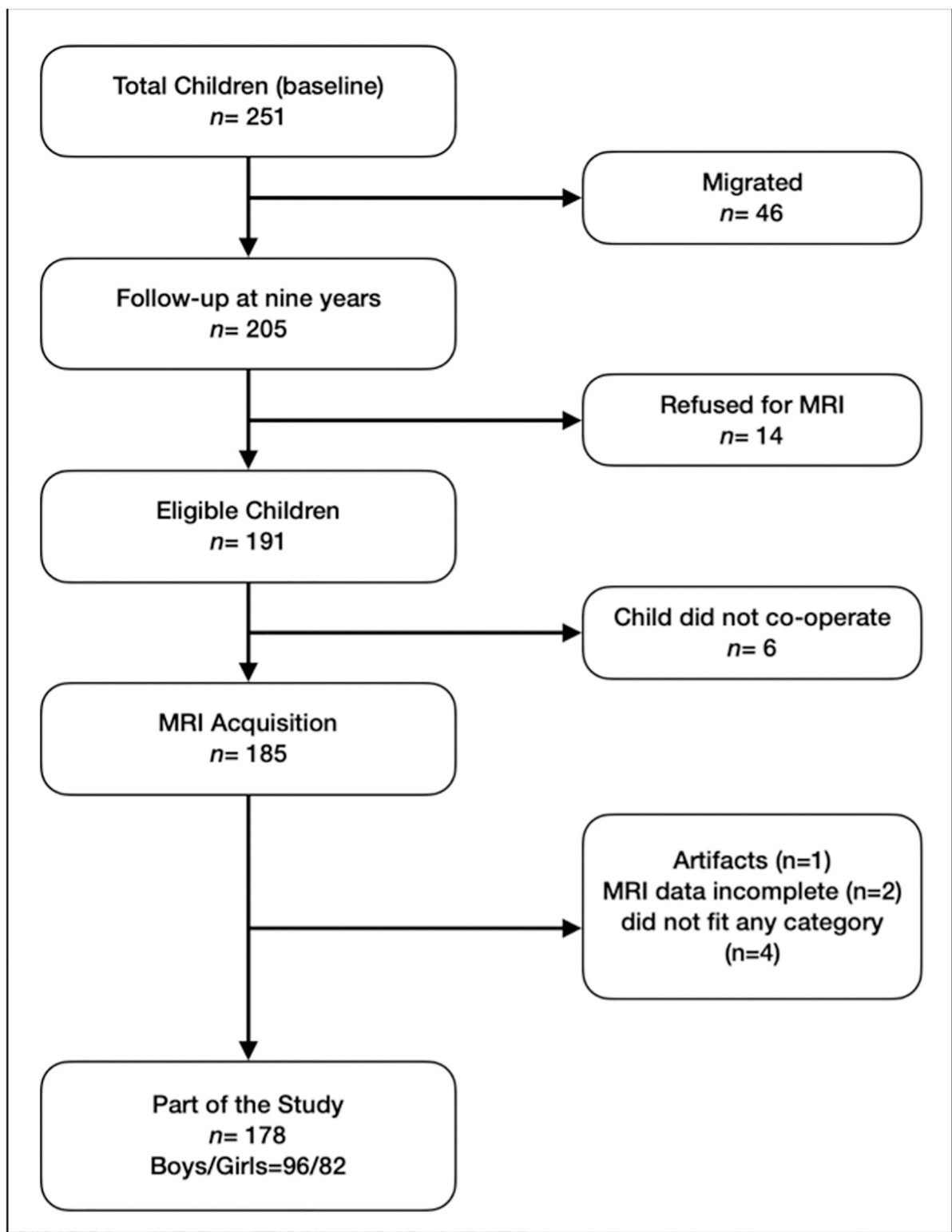

**Fig 1. Flowchart illustrating the enrollment process for the current study.**

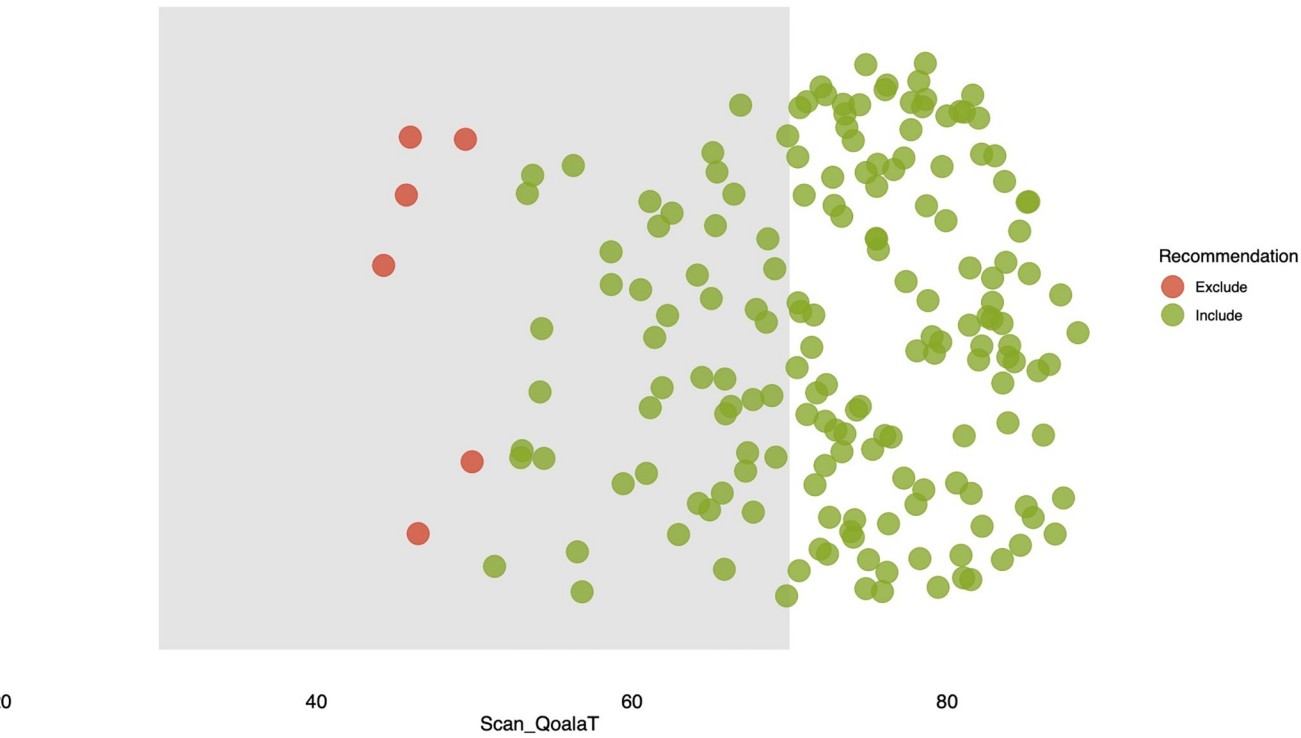

**Fig 2. Qoala-T graph displaying the number of included and excluded predictions.**

information and movement artifacts. Four children did not fit into either of the four growth groups (Groups AS–NS) described here and were excluded from analysis.

Utilizing the Qoala-T prediction model on the remaining scans, we derived Qoala-T scores, which serve as indicators of scan quality. Fig 2 illustrates the Qoala-T graph, offering recommendation on the inclusion and exclusion of scans. Out of the total scans, 121 were recommended for inclusion without the necessity of further visual quality control (QC), while an additional 51 scans required supplementary visual QC for inclusion. Notably, six scans though slated for exclusion were found to be eligible onr a secondary visual QC. Information detailing the scores of each scan, along with corresponding recommendations, is available in the supplementary information.

Subsequent visual QC assessments of the recommended included and excluded scans revealed no discrepancies. Ultimately, a subsample of 178 children was used for our analysis, of which 96 had never experienced stunting (Group NS), 31 had experienced stunting at the age of two with a catch-up at the age of five (Group S2N5), 29 had experienced stunting at the ages of two and five with a catch-up at the age of nine (Group S5N9), and 22 had experienced stunting at the ages of two, five, and nine (Group AS) (Fig 1). The demographic information for the research population is explained in Table 1.

## Brain volumetry

Differences between the brain region volumes between groups have been depicted in Table 2. The total brain volume decreased with the number of years stunted mean (95% CI): 1070455 mm$^3$ (1055101 mm$^3$-1085809 mm$^3$) vs 1048764 mm$^3$ (1022428 mm$^3$-1075101 mm$^3$) vs 1040409 mm$^3$ (1012365 mm$^3$-1068453 mm$^3$) vs 1026864 mm$^3$ (995656 mm$^3$-1058071 mm$^3$)

**Table 1. Demographics of the analysed cohort.**

| Characteristics | NS | S2N5 | S5N9 | AS |
|---|---|---|---|---|
| Age in Years; Mean (Range) | 9.38 (9–11) | 9.68 (9–11) | 9.86 (9–11) | 9.68 (9–11) |
| Height, Cm; Mean (Range) | 132.04 (131.1–133) | 129.08 (128–130.2) | 123.62 (122.8–124.5) | 119.07 (118–120.1) |
| Weight, Kg; Mean (Range) | 27.77 (26.4–29.1) | 25.71 (24.38–27) | 21.37 (20.3–22.5) | 20.02 (18.8–21.2) |
| Sex | | | | |
| Boys; N(%) | 36 (37.5%) | 16 (51.61%) | 18 (62.07%) | 12 (54.55%) |
| Girls; N(%) | 60 (62.5%) | 15 (48.39%) | 11 (37.93%) | 10 (45.45%) |
| Handedness | | | | |
| Right; N(%) | 95 (98.85%) | 31 (100%) | 29 (100%) | 22 (100%) |
| Left; N(%) | 1 (1.15%) | - | - | - |

Comparisons were done using ANOVA between the children groups. The mean of the characteristics within the different groups were estimated with 95% confidence interval. Abbreviations: NS–Never stunted, S2N5—Stunted at 2 years and caught up by 5 years, S5N9—Stunted at 2 and 5 years and caught up by 9 years, AS–Always Stunted

for children in Groups NS-AS respectively with AS showing significant lower total brain volume compared to NS ($p = 0.01$).

There was a decrease in the overall sub cortical gray matter volume across the groups NS-AS (54466 mm$^3$ vs 53303 mm$^3$ vs 53131 mm$^3$ vs 51917 mm$^3$ respectively), exhibiting statistical differences between Group NS and Group AS, ($p = 0.01$). Decreased volumes were seen in overall white matter within the bilateral hemispheres, across the groups, [Left, 19144 mm$^3$ vs 18911 mm$^3$ vs 17812 mm$^3$ vs 17018 mm$^3$, $p = 0.05$; Right, 19012 mm$^3$ vs 18533 mm$^3$ vs 17705 mm$^3$ vs 17017 mm$^3$, $p = 0.04$ NS-AS respectively] exhibiting a near statistical significance between Group NS and Group AS.

## Cerebellum

Bilateral cerebellar white matter volumes showed a significant difference between Groups NS and AS [Left, 12092 mm$^3$ Vs 11321 mm$^3$ respectively, $p = 0.03$; Right, 11749 mm$^3$ Vs 11073 mm$^3$ respectively, $p = 0.04$] and showed dose response with catch-up growth as well. Bilateral cerebellar cortices showed a significant difference between Groups NS and AS [Left, 53709 mm$^3$ Vs 50702 mm$^3$ respectively, $p = 0.001$; Right, 54071 mm$^3$ Vs 50462 mm$^3$ respectively, $p = 0.0003$], but a similar dose response with catch-up growth was not seen. (see Table 2). In addition to this, there was a moderate effect of stunting on bilateral regions of the cerebellum within all the groups [Left, $\eta^2 = 0.07$; Right, $\eta^2 = 0.08$]

## Frontal region

On analysing the volume and thickness of multiple regions within the frontal lobe including the orbito-frontal region, caudal middle frontal region, rostral middle frontal region and the frontal pole region, there was no significance found between groups.

## Parietal region

Evaluating inferior parietal region, the volumes of the right inferior parietal side exhibited statistical significance (17542 mm$^3$ vs 16318 mm$^3$ respectively; $p = 0.01$), between Group NS and Group S2N5, and (17542 mm$^3$ vs 16048 mm$^3$ respectively, $p = 0.004$), between Group NS and Group S5N9. There was also a moderate effect of stunting in this region, ($\eta^2$) = 0.08. Within the superior parietal regions, similar differences were not seen between AS and NS groups.

**Table 2. Morphometric analysis of brain volumes exhibiting significance due to stunting.**

| | | Group NS | Group S2N5 | P-Value | Group S5N9 | P-Value | Group AS | P-Value | Effect-Size |
|---|---|---|---|---|---|---|---|---|---|
| Total Brain | | 1070455 (1055101–1085809) | 1048764 (1022428–1075101) | 0·2 | 1040409 (1012365–1068453) | 0·1 | 1026864 (995656·8–1058071) | **0·01** | 0·04 |
| Sub Cortical Gray Matter | | 54466·43 (53651·24–55281·62) | 53302·57 (51904·33–54700·8) | 0·14 | 53130·86 (51641·96–54619·77) | 0·22 | 51917·36 (50260·53–53574·19) | **0·01** | 0·05 |
| Unsegmented White Matter | L | 19144·99 (18398·28–19891·7) | 18911·26 (17630·48–20192·03) | 0·82 | 17812·01 (16448·19–19175·83) | 0·15 | 17108·38 (15590·74–18626·02) | **0·05** | 0·04 |
| | R | 19012·37 (18297·02–19727·72) | 18532·93 (17305·93–19759·92) | 0·56 | 17705·19 (16398·63–19011·74) | 0·15 | 17017·39 (15563·48–18471·3) | **0·04** | 0·04 |
| Cerebellum White Matter | L | 12092·6 (11816·95–12368·24) | 11861·61 (11388·82–12334·4) | 0·46 | 11543·64 (11040·19–12047·09) | 0·08 | 11321·01 (10760·77–11881·24) | **0·03** | 0·04 |
| | R | 11748·84 (11499·79–11997·89) | 11455·43 (11028·26–11882·61) | 0·25 | 11373·53 (10918·65–11828·41) | 0·1 | 11072·71 (10566·53–11578·89) | **0·04** | 0·04 |
| Cerebellum Cortex | L | 53709·02 (52873·5–54544·53) | 51889·19 (50456·1–53322·28) | **0·02\*** | 52194·36 (50668·34–53720·38) | 0·13* | 50702·14 (49004·01–52400·27) | **0·001\*** | 0·07 |
| | R | 54071·97 (53212·6–54931·34) | 52385·93 (50911·92–53859·94) | **0·04\*** | 52791·24 (51221·64–54360·83) | 0·2 | 50462·74 (48716·12–52209·36) | **0·0003\*** | 0·08 |
| **Parietal Volumes** | | | | | | | | | |
| Superior Parietal | L | 15194·23 (14783·24–15605·22) | 15234·37 (14529·43–15939·31) | 0·82 | 14984·81 (14234·15–15735·46) | 0·54 | 14452·23 (13616·91–15287·54) | 0·09 | 0·02 |
| | R | 15100·69 (14706·01–15495·37) | 14786·24 (14109·27–15463·21) | 0·58 | 14690·85 (13969·98–15411·72) | 0·35 | 14467·44 (13665·27–15269·61) | 0·16 | 0·01 |
| Inferior Parietal | L | 14390·27 (13993·43–14787·11) | 13633·95 (12953·29–14314·62) | 0·09 | 13899·56 (13174·75–14624·36) | 0·41 | 14021·59 (13215·05–14828·14) | 0·48 | 0·02 |
| | R | 17542·87 (17121·88–17963·86) | 16318·74 (15596·65–17040·83) | **0·01\*** | 16047·54 (15278·63–16816·46) | **0·004\*** | 17045·53 (16189·89–17901·16) | 0·36 | 0·08 |
| **Occipital Volumes** | | | | | | | | | |
| Lateral Occipital | L | 13457·53 (13114·8–13800·25) | 12915·07 (12327·23–13502·92) | 0·13 | 12267·33 (11641·37–12893·3) | **0·01\*** | 12558·66 (11862·09–13255·22) | **0·03\*** | 0·07 |
| | R | 13860·09 (13499·08–14221·09) | 13308·09 (12688·89–13927·29) | 0·17 | 12129·89 (11470·54–12789·25) | **0\*** | 13074·73 (12341·01–13808·45) | **0·05\*** | 0·11 |
| **Sub Cortical Gray Volumes** | | | | | | | | | |
| Putamen | L | 4852·13 (4730·26–4974·01) | 4791·82 (4582·78–5000·86) | 0·56 | 4841·47 (4618·87–5064·06) | 0·97 | 4638·78 (4391·08–4886·48) | 0·22 | 0·01 |
| | R | 4937·32 (4827·64–5047) | 4893·61 (4705·49–5081·74) | 0·7 | 4914·87 (4714·54–5115·2) | 0·93 | 4882·9 (4659·98–5105·82) | 0·84 | 0·001 |
| Pallidum | L | 1790·43 (1744·7–1836·17) | 1783·74 (1705·29–1862·19) | 0·97 | 1780·76 (1697·22–1864·29) | 0·89 | 1722·22 (1629·26–1815·17) | 0·2 | 0·01 |
| | R | 1705·48 (1664·24–1746·72) | 1656·38 (1585·65–1727·12) | 0·25 | 1655·94 (1580·62–1731·26) | 0·37 | 1580·9 (1497·09–1664·72) | **0·01** | 0·04 |
| Caudate | L | 3488·37 (3403·64–3573·1) | 3367·54 (3222·21–3512·87) | 0·09 | 3319·84 (3165·09–3474·6) | 0·07 | 3267·3 (3095·09–3439·51) | **0·02** | 0·04 |
| | R | 3572·96 (3491·34–3654·58) | 3535·31 (3395·31–3675·31) | 0·45 | 3422·86 (3273·78–3571·94) | 0·1 | 3383·25 (3217·26–3549·14) | **0·03** | 0·03 |
| Thalamus | L | 6594·33 (6468·93–6719·74) | 6515·21 (6300·11–6730·31) | 0·6 | 6554·19 (6325·14–6783·24) | 0·89 | 6296·6 (6041·72–6551·48) | 0·06 | 0·02 |
| | R | 6559 (6441·78–6676·22) | 6421·18 (6220·13–6622·23) | 0·28 | 6435·24 (6221·15–6649·33) | 0·57 | 6156·89 (5918·66–6395·13) | **0·01** | 0·05 |
| **Corpus Callosum (CC)** | | | | | | | | | |
| CC Posterior | | 789·38 (764·92–813·84) | 756·08 (714·12–798·04) | 0·31 | 748·16 (703·48–792·84) | 0·25 | 686·24 (636·52–735·97) | **0·002\*** | 0·07 |

*(Continued)*

**Table 2.** (Continued)

| | Group NS | Group S2N5 | P-Value | Group S5N9 | P-Value | Group AS | P-Value | Effect-Size |
|---|---|---|---|---|---|---|---|---|
| CC Mid Posterior | 439·2 (423·48–454·94) | 430·24 (403·25–457·23) | 0·74 | 447.51 (418·77–476·25) | 0·39 | 398·12 (366·13–430·1) | 0·06 | 0·04 |

ANOVA was done to compare the mean volume and thickness values of different brain regions between the groups by keeping age and sex as co-variates, with 95% confidence interval. The effect size of stunting on the different brain regions was determined using Cohen's d. Abbreviations: NS–Never stunted, S2N5—Stunted at 2 years and caught up by 5 years, S5N9—Stunted at 2 and 5 years and caught up by 9 years, AS–Always stunted, L = Left, R = Right. Bolded values have significant p-values. * represents a medium effect of stunting on the particular region while ** represents a high effect.

### Occipital region

Evaluating the occipital region, the bilateral lateral occipital volumes [Left, 13458 mm$^3$ vs 12559 mm$^3$ respectively, *p = 0·03*; Right, 13860 mm$^3$ vs 13075 mm$^3$ respectively, *p = 0·05*] were significantly smaller in the AS when compared to NS (Table 2). Similarly, there was also statistical difference visible between the Group NS and Group S5N9. There was also a moderate effect of stunting visible, Left ($\eta^2$) = 0·07, Right($\eta^2$) = 0·11.

### Temporal region

Evaluation of multiple regions of temporal lobe did not yield any significant results.

### Subcortical gray matter

Table 2 and Fig 3 provide volume variations in the subcortical brain areas among different child groups. Areas of right pallidum (1705 mm$^3$ vs 1581 mm$^3$ respectively, *p* = 0·01) left caudate (3488 mm$^3$ vs 3267 mm$^3$ respectively, *p = 0·02*), right caudate (3573 mm$^3$ vs 3383 mm$^3$ respectively, *p = 0·03*), and right thalamus (6559 mm$^3$ vs 6157 mm$^3$ respectively, *p = 0·01*) displayed statistical difference between Group NS and Group AS.

### Corpus callosum

Among the regions within the Corpus Callosum, only the posterior (789 mm$^3$ vs 686 mm$^3$ respectively, *p = 0·002*) region showed statistical significance between Group NS and Group AS. There was also a moderate effect of stunting witnessed in the posterior region alone [$\eta^2$ = 0·07].

### Discussion

The present study evaluated brain volumetry associations in children persistently stunted in comparison with those never stunted and those who caught up in growth. AS children had smaller total brain volumes, total subcortical gray matter volumes, bilateral cerebellum, and posterior corpus callosum volumes when compared to those NS. Regional cortical brain analysis showed lower bilateral lateral occipital volumes, right pallidum, bilateral caudate, and right thalamus volumes in the AS. Dose response was seen with the number of years of stunting in total brain volume, total subcortical gray matter volumes, posterior corpus callosum volume, and bilateral cerebellar white matter. To the best of our knowledge, this is the first neuroimaging analysis to investigate the effects of persistent childhood stunting, and catch-up growth on brain volumes in school-going children.

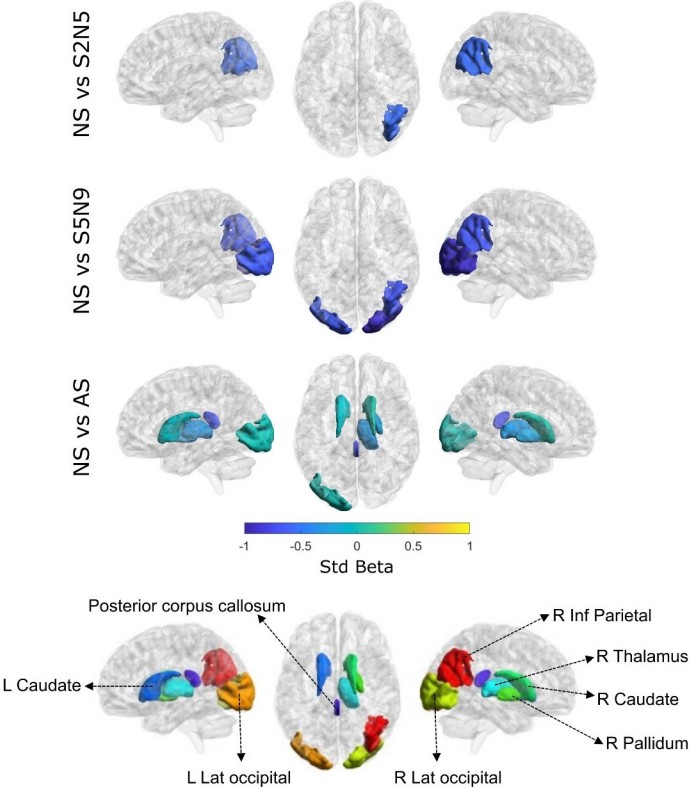

**Fig 3. Study groups showing significant differences in subcortical brain regions.** Standardized beta values indicating significant differences among subcortical and cortical brain regions are shown here. Abbreviations: NS–Never stunted, S2N5—Stunted at 2 years and caught up by 5 years, S5N9—Stunted at 2 and 5 years and caught up by 9 years, AS–Always stunted. Only AS (Always Stunted) children exhibited reduced subcortical brain volumes specifically in the right caudate nucleus, left caudate nucleus, right pallidum and right thalamus when compared to NS (Never Stunted) children.

In a previous report from the same cohort, children who remained persistently stunted had a significant reduction in verbal and total cognition scores, by 4·6 points which corresponded to a 0·3 standard deviation difference compared to those never stunted, and are consistent with the previous literature [6, 21]. The first 1000 days are crucial for brain development with respect to neurogenesis, synaptogenesis, and myelination, and early childhood chronic malnutrition manifesting as stunting can affect this process, causing persistent effects including cognitive and behavioural deficits. The current analysis adds to the existing literature that children persistently stunted showed volume loss at different brain levels involving overall brain volumes, networking/connecting centres (thalamus, basal ganglia, callosum, cerebellum) and visual processing area of lateral occipital cortex.

There are very few studies evaluating neuroimaging findings in children with stunting, especially in a birth cohort follow-up setting, highlighting this as a research gap [10, 11]. According to a recent scoping analysis by Ayaz, volume loss as evidenced by cerebral atrophy and dilated ventricles is the most frequent finding in malnourished children [22]. This can represent overall suboptimal brain development in the background of poor nutrition, especially in the first 1000 days. In the current study also, total brain volume at nine years was found to be smaller in children with persistent stunting when compared to healthy children and children with catch-up growth. Multi-modal MRI brain analysis of the Philadelphia

Neurodevelopmental Cohort (PNC) evaluated brain maturation and found that lower socio-economic position in childhood was associated with lower gray-matter density [23]. In contrast, Malawian children who had severe acute malnutrition were noticed to have no difference in radiologist-checked brain abnormalities when compared with peers despite being cognitively poor [24]. Differing MRI Brain outcome analyses and sub-optimal standardisation might explain some of these differences in literature. The current study utilised standardised practices for 3T MRI acquisition and volumetry was performed using Freesurfer™ software.

The present study also addressed whether catch-up growth identified by anthropometric markers can be associated with brain volumetric changes. In our analysis, total brain volumes, total subcortical gray matter volumes, bilateral cerebellar white matter volumes, and posterior corpus callosum showed a dose-response volume difference with the number of years of stunting and correspondingly catch-up. The larger MAL-ED study had shown that catch-up growth was seen in stunted children probably by a contribution of nutritional and environmental factors [25]. Longitudinal structural MRI studies conducted on eight-year-old children born preterm with very low birth weight in the Norwegian Mother and Child Cohort Study, showed no relation between catch-up growth and brain volume improvement [26]. However, smaller volumes of corpus callosum, right pallidum, and right thalamus were observed in the very low birth weight group analogous to the current study. MRI scans and cognition data obtained from preterm children enrolled in the Norwich nutritional study showed that total brain volumes were higher in those receiving high-nutrient diet when compared to those on standard diet showing the nutritional impact on brain reorganization [27].

Basal ganglia including caudate nucleus and putamen are involved in complex data selection, networking, and relay as well as facilitation of purposeful movements with inhibition of interfering activities [28]. What is highlighted recently is the increased integration of basal ganglia system with internal brain structures of cortex and cerebellum and external cues of sensation and inputs [28]. The high energy need of these intense complex sub-cortical structures can make them vulnerable to nutritional threats such as stunting as seen in this study. The Norwich Nutritional study had also shown that caudate volumes were larger in high-nutrient group males similar to our study [27].

Cerebellar cortical and white matter volumes were found to be smaller in children who were always stunted. In addition, cerebellar white matter showed a dose response by improvement with catch-up growth. Paucity of studies evaluating cerebellum in the context of stunting limits a complete comparative analysis. In a follow-up study of children born small for gestational age, both cerebral and cerebellar gray and white matter volumes were smaller at four-seven years of age compared to those born with normal birth weight [29]. It might be possible that chronic malnutrition can deleteriously affect neurogenesis resulting in a smaller number of gray matter cells and in turn lower gray matter volume in both cerebrum and cerebellum. The cerebellum is known as a network/integrating/co-ordination centre and influences motor co-ordination, oculomotor control, speech clarity, language, cognition, and attention and in this current cohort, children AS had both lower verbal and total cognition scores when compared to NS [6, 30]. Both basal ganglia and cerebellum are complex brain structures, functions of which are being unravelled and similar pathophysiology may be involved to affect these systems in chronic malnutrition as seen in persistent stunting. Further studies can explore underlying pathophysiology mechanism to find any possible rectification.

We also observed smaller volumes of the posterior corpus callosum in children always stunted when compared to those not stunted and those who had catch-up growth. Like the cerebellum, few studies have explored corpus callosum volumes in children presented with stunting. Smaller corpus callosum volumes were found in school-age children born with very low birth weight [26]. Corpus callosum connecting cerebral hemispheres acts to integrate

information from both sides effectively and efficiently. The posterior part of the corpus callosum integrates occipital lobes where vision is localised as well as posterior parietal regions where cognition association areas are located [31]. In the current cohort, lateral occipital cortex, the visual processing hub is also smaller in persistently stunted children indicating involvement of higher order association areas as well. In a large meta-analysis, early childhood adversity such as low socio-economic position was found to be associated with lower executive functions in children and further evaluation can explore the linkage between childhood adversity, brain association area structures and higher order cognition including executive function [32].

The current study is unique in brain volume analysis in children persistently stunted and those caught up in growth. Main strengths of this study are the relatively large Indian birth cohort that has been followed up over the years with good early childhood data, standardised length assessment with inbuilt quality control, and 3T MRI scan at nine years follow-up. Nevertheless, there are certain limitations that need to be considered. The current study had a sample size of 178 and was limited to Vellore in south India. The Covid-19 epidemic in 2020–2021 caused the MRI acquisition to be stopped during the follow-up research. Age disparities emerged as a result during the nine-year follow-up and thus all analyses were adjusted for age.

## Conclusion

To the best of our knowledge, this first neuroimaging analysis to investigate the effects of persistent childhood stunting and catch-up growth on brain volumetry indicates impairment at different brain levels involving overall brain volumes, networking/connecting centres (thalamus, basal ganglia, callosum, cerebellum) and visual processing area of lateral occipital cortex. Future research can explore underlying pathophysiology as well as prospective neuroimaging studies to validate/extend current findings. Understanding stunting effects from a biological and neurological perspective can help to develop supportive relevant interventions in future.

## Acknowledgments

The authors thank the participants, their families and staff of the MAL-ED Network project.

## Author Contributions

**Conceptualization:** Beena Koshy, Venkata Raghava Mohan, Gagandeep Kang.

**Data curation:** Beena Koshy, Vedha Viyas Thilagarajan, Manikandan Srinivasan, Venkata Raghava Mohan.

**Formal analysis:** Beena Koshy, Vedha Viyas Thilagarajan, Samuel Berkins, Arpan Banerjee.

**Funding acquisition:** Beena Koshy, Gagandeep Kang.

**Investigation:** Beena Koshy, Samuel Berkins, Roshan S. Livingstone, Anitha Jasper.

**Methodology:** Beena Koshy, Arpan Banerjee, Manikandan Srinivasan, Roshan S. Livingstone, Venkata Raghava Mohan, Rebecca Scharf, Anitha Jasper, Gagandeep Kang.

**Project administration:** Beena Koshy, Samuel Berkins, Roshan S. Livingstone, Anitha Jasper.

**Resources:** Beena Koshy, Venkata Raghava Mohan, Gagandeep Kang.

**Supervision:** Beena Koshy, Arpan Banerjee, Roshan S. Livingstone, Venkata Raghava Mohan, Rebecca Scharf, Gagandeep Kang.

**Validation:** Beena Koshy.

**Visualization:** Beena Koshy, Vedha Viyas Thilagarajan, Venkata Raghava Mohan.

**Writing – original draft:** Beena Koshy, Vedha Viyas Thilagarajan, Samuel Berkins, Arpan Banerjee, Manikandan Srinivasan, Roshan S. Livingstone, Venkata Raghava Mohan, Rebecca Scharf, Anitha Jasper, Gagandeep Kang.

**Writing – review & editing:** Beena Koshy, Vedha Viyas Thilagarajan, Samuel Berkins, Arpan Banerjee, Manikandan Srinivasan, Roshan S. Livingstone, Venkata Raghava Mohan, Rebecca Scharf, Anitha Jasper, Gagandeep Kang.

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
