## [Decision Letter · Decision Letter 0]

12 Aug 2024

PONE-D-24-24149Childhood brain morphometry in children with persistent stunting and catch-up growthPLOS ONE

Dear Dr. Koshy,

Thank you for submitting your manuscript to PLOS ONE. After careful consideration, we feel that it has merit but does not fully meet PLOS ONE’s publication criteria as it currently stands. Therefore, we invite you to submit a revised version of the manuscript that addresses the points raised during the review process.

We look forward to receiving your revised manuscript.

Kind regards,

Cota Navin Gupta

Academic Editor

PLOS ONE

Journal Requirements:

2. Thank you for stating the following financial disclosure: a. Bill and Melinda Gates Foundation

The Etiology, Risk Factors and Interactions of Enteric Infections and Malnutrition and the Consequence for Child Health and Development Project (MAL-ED) is carried out as a collaborative project supported by the Bill and Melinda Gates Foundation, the Foundation for the NIH and the National Institutes of Health/Fogarty International Center (Grant number – OPP 47075)

b. DBT-Wellcome Trust India Alliance 

The 9-year follow-up of the Mal-ed India cohort was supported by an Intermediate clinical and public health research fellowship awarded by the DBT/Wellcome Trust India Alliance to Dr. BK. (Fellowship grant number IA/CPHI/19/1/504611)  

3. In the online submission form, you indicated that All MAL-ED data till 5 years are available on www.clinEpiDB.

Further data can be shared by corresponding author on reasonable request. 

Additional Editor Comments:

Interesting dataset you have collected. However you are requested to respond to queries raised by me (below) and reviewers about your study. In the revised manuscript please highlight the changes made in a different colour (i.e. for each reviewer and Editor) and in reply to reviewers document mention the line numbers of revised manuscript where you have addressed the issue…We look forward to receiving your replies/revisions..Good luck

1)In Abstract- Please specify the methodology as suggested by reviewer 4

Also In Introduction- Why Freesurfer was used to study morphometry..add your motivation?

2)Why Desikan killiany atlas was used...Any reason or reference?

3)Studies have shown sex and age differences as well as sex by age interaction for children.. See below references

https://pubmed.ncbi.nlm.nih.gov/23500670/

https://doi.org/10.1093/cercor/11.6.552

I am not sure if sex by age interaction was considered in your model?

4)You seem to have missed some latest studies. If relevant discuss these in introduction or discussion sections as appropriate in context of their results.

https://www.sciencedirect.com/science/article/pii/S0361923023002721

https://www.ncbi.nlm.nih.gov/pmc/articles/PMC10964341/

Reviewers' comments:

Reviewer's Responses to Questions

**Comments to the Author**

1. Is the manuscript technically sound, and do the data support the conclusions?

Reviewer #1: Yes

Reviewer #2: Yes

Reviewer #3: Partly

Reviewer #4: Partly

2. Has the statistical analysis been performed appropriately and rigorously? 

Reviewer #1: Yes

Reviewer #2: I Don't Know

Reviewer #3: I Don't Know

Reviewer #4: I Don't Know

3. Have the authors made all data underlying the findings in their manuscript fully available?

Reviewer #1: No

Reviewer #2: Yes

Reviewer #3: No

Reviewer #4: No

4. Is the manuscript presented in an intelligible fashion and written in standard English?

Reviewer #1: Yes

Reviewer #2: Yes

Reviewer #3: No

Reviewer #4: Yes

5. Review Comments to the Author

Reviewer #1: This work studies early childhood stunting effects. The authors highlighted that this is the first neuroimaging analysis to investigate the effects of childhood stunting. Specifically, this study analyses correlations between childhood stunting occurrence and multiple covariants, including total brain volume, subcortical volume, bilateral cerebellar white matter, posterior corpus callosum, etc. Statistical analyses including p-value, are conducted, categorized by different regions.

I think this work is coherent and supportive, and can provide useful references for future research on the reasonings and trends of stunting effects.

Reviewer #2: This study provides valuable insights into the long-term neurological impacts of early childhood stunting and subsequent catch-up growth, an area with limited prior data.

While this study offers important insights, there are several limitations to consider. The sample size of 178 children, though informative, may not fully capture the variability within the broader population.

Additionally, the study is geographically limited to Vellore, south India, which might affect the generalizability of the findings to other regions or populations.

The cross-sectional nature of the neuroimaging analysis at nine years of age does not allow for the assessment of brain development trajectories over time.

Furthermore, potential confounding factors such as socioeconomic status, educational environment, and genetic predispositions were not thoroughly addressed, which could influence the observed outcomes.

Most of the references are clubbed. Its better to use one ciation at the end of one line.

Reviewer #3: 1.1. Current literature has established that handedness is associated with differences in brain morphology. Given that the current study has only included one left-handed participant, rather than including that participant, it may be worth removing that participant.

1.2. From the literature review/introduction, I understand that there are very few work looking at childhood stunting and brain morphometry. However, it would have been beneficial to include a broader range of knowledge, such as studies examining malnutrition (which is closely related to stunting) and brain morphometry, or research on childhood stunting and its effects on other brain modalities, such as fMRI.

1.3. The study mentions addressing the research gap of limited data on the effects of early childhood stunting and catch-up growth on brain morphometry. However, the conclusion primarily emphasized the differences between the 'Never Stunted' and 'Always Stunted' children. The 'caught-up' groups were not discussed in detail, and there was limited interpretation of the findings related to these groups.

2.1. Aside from age and gender, total intracranial volume is also an important covariate for volumetric analyses, which I don't think has been considered in the current study. In addition, I am curious whether the present study has considered the child's awake/asleep state during the MRI scan as a covariate.

2.2. I understand that visual checks have been carried out visually, but I am wondering if post segmentation checks have been performed (e.g., using Qoala-T on the Freesurfer output) as the study has not mentioned and it will be important information.

3. Authors have stated that some restrictions will apply.

4. There are some punctuation and grammatical errors, which may be a result of the long sentences. For example - "In a previous report from the same cohort, children who remained persistently stunted had a significant reduction in verbal, and total cognition scores by 4·6 points which corresponded to a 0·3 standard deviation difference compared to those never stunted, and is consistent with the previous literature [6, 17]." The comma is misplaced, and there are grammatical errors: "In a previous report from the same cohort, children who remained persistently stunted had significant reductions in verbal and total cognition scores, by 4·6 points which corresponded to a 0·3 standard deviation difference compared to those children who have never stunted, and are consistent with the previous literature [6, 17]."

5. I was curious on why the study looked till 9 years old, and not an older age since the study was interested in the catch-up growth on brain morphometry (for example, 12 years old is an age at which brain volume is suggested to continue growing till).

Reviewer #4: The work by Koshy et al to understand the morphometric changes associated with stunting and catch-up growth is interesting with a larger Indian cohort from a small town. Though I liked the approach but the study needs considerable improvement from the analysis point of view.

MAJOR

Abstract- Please specify the methodology used in the method section to let readers understand what was done before they understand the finding. Please specify multiple comparison etc here

Introduction-

1. Please define stunting and the criteria to identify

2. The statement 1000days = 2 years is bit confusing.

Method

3. Every study should be an independent read. So please briefly mention the cohort details in main manuscript or in supplement.

4. Please mention the age at which the childrens were scanned. Additionally, provide details on How many children needed Triclofos- Please mention.

5. Are the Regional Brain volumes normalised. Most of regional level results will dissapear if Total brain volume is regressed.

6. How was the Time of MRI aquisition adjusted.

Result

Please provide all the details of ANNOVA analysis for each group. Please adjust the time of aquistion, TIV and results will get shifted from present. This will provide a robust picture of the analysis.

MINOR

(1) Writing needs to be improved through out the manuscript at several places- starting from abstract to discussion- For example- Amongst 251 children from the overall cohort, 178 children with a mean age of 9.54

were considered for further analysis.- It sounds like the volumetric parameter is assessed longitudinally.

Please write the message in each Paragraphs clearly and directly. For example Paragraph 3 shows Neuroimaging is expensive which I agree, but authors themselves have used neuroimaging rather than any alternative.

6. PLOS authors have the option to publish the peer review history of their article (what does this mean?). If published, this will include your full peer review and any attached files.

Reviewer #1: No

Reviewer #2: **Yes: **Apurba Patra

Reviewer #3: No

Reviewer #4: **Yes: **Rajan Kashyap

---

## [Author Response · Author response to Decision Letter 0]

17 Sep 2024

To,

The Editor, Reviewers,

PLOS ONE

Date: 23 August 2024

Subject: Addressing Editor/Reviewer Comments for PONE-D-24-24149

Dear Editor, Reviewers,

Many thanks to you for your helpful feedback. We have addressed all your concerns and changes are highlighted in the manuscript. Responses to concerns are highlighted in red in this document. Let us know if you have any further queries.

Thank you. 

Kind regards, 

Authors of PONE-D-24-24149

1)In Abstract- Please specify the methodology as suggested by reviewer 4

Answer: We have added the methodology used in the study, under the “Methods” section in the “Abstract”. 

Also In Introduction- Why Freesurfer was used to study morphometry..add your motivation?

Answer: We have added our motivation for using Freesurfer in this study in the MRI Image Acquisition and Processing paragraph under the “Materials and Methods” section. Freesurfer has been chosen for this study due to its widespread acceptance as a standard for brain volumetric studies and its validation for use in children aged 4 to 11 years. (258-260)

2)Why Desikan killiany atlas was used...Any reason or reference?

Answer: We have added the reason for using Desikan Killiany atlas in the MRI Image Acquisition and Processing paragraph under the “Materials and Methods” section. The selection of the Desikan-Killiany atlas for this study was based on its consistent representation of atrophy levels and accurate depiction of the macroscopic anatomy of gyri and sulci. This made it a preferred choice over other atlases for examining both healthy children and those with stunted growth. (266-269)

3)Studies have shown sex and age differences as well as sex by age interaction for children.. See below references

https://pubmed.ncbi.nlm.nih.gov/23500670/

https://doi.org/10.1093/cercor/11.6.552

I am not sure if sex by age interaction was considered in your model?

Answer: Sex by age has not been considered in our analyses as most of the scans were done at the Age of 9 and only a few were done after a year, due to disruption caused by Covid-19. Therefore, we had adjusted for sex and age separately in our model.

4)You seem to have missed some latest studies. If relevant discuss these in introduction or discussion sections as appropriate in context of their results.

https://www.sciencedirect.com/science/article/pii/S0361923023002721

https://www.ncbi.nlm.nih.gov/pmc/articles/PMC10964341/

Answer: Thank you for suggesting the above studies. The above studies have focused on studying brain morphometry related to Chronic Hepatic Encephalopathy and Pediatric Hydrocephalus, while the current study lays emphasis on studying the effect of early childhood stunting on brain morphometry. In relation to that, we have reviewed literatures that had focused on outcomes related to childhood stunting.

Reviewer #1: This work studies early childhood stunting effects. The authors highlighted that this is the first neuroimaging analysis to investigate the effects of childhood stunting. Specifically, this study analyses correlations between childhood stunting occurrence and multiple covariants, including total brain volume, subcortical volume, bilateral cerebellar white matter, posterior corpus callosum, etc. Statistical analyses including p-value, are conducted, categorized by different regions.

I think this work is coherent and supportive and can provide useful references for future research on the reasonings and trends of stunting effects.

Answer: Thank you for your encouraging comments.

Reviewer #2: This study provides valuable insights into the long-term neurological impacts of early childhood stunting and subsequent catch-up growth, an area with limited prior data.

While this study offers important insights, there are several limitations to consider. The sample size of 178 children, though informative, may not fully capture the variability within the broader population.

Additionally, the study is geographically limited to Vellore, south India, which might affect the generalizability of the findings to other regions or populations.

Answer: Thank you. We agree with limitations suggested and have included these in limitations in the study (lines 540-41).

The cross-sectional nature of the neuroimaging analysis at nine years of age does not allow for the assessment of brain development trajectories over time.

Answer: Thank you. We agree. In the same birth cohort, we have recently completed 12-year neuroimaging analysis, which will be further evaluated for a possible trajectory evaluation.

Furthermore, potential confounding factors such as socioeconomic status, educational environment, and genetic predispositions were not thoroughly addressed, which could influence the observed outcomes.

Answer: Thank you. We agree with the observation. We have another publication from the same cohort “Koshy B, Srinivasan M, Gopalakrishnan S, Mohan VR, Scharf R, Murray-Kolb L, et al. Are early childhood stunting and catch-up growth associated with school age cognition?-Evidence from an Indian birth cohort. PloS one. 2022;17(3):e0264010. Epub 20220302. doi: 10.1371/journal.pone.0264010. PubMed PMID: 35235588; PubMed Central PMCID: PMCPMC8890627.”, which has evaluated these factors. Persistent stunting had an effect on both verbal and performance quotient by around 5 points even after correction with SES and maternal cognition. The current paper is a follow-up analysis of the previous published analysis.

Most of the references are clubbed. Its better to use one citation at the end of one line.

Answer: Thank you. We have tried to give specific references now.

Reviewer #3: 1.1. Current literature has established that handedness is associated with differences in brain morphology. Given that the current study has only included one left-handed participant, rather than including that participant, it may be worth removing that participant.

Answer: Thank you for your comment. We agree that handedness play a role in differences in brain morphology based on current literature. But from our field experience, we have noted that many study children, who were born left-handed, had been trained to be right-handed by their parents. Therefore, we chose not to add handedness as a confounding factor between the association of early childhood stunting and brain morphometry, and also keep the corresponding participant in the analysis. 

1.2. From the literature review/introduction, I understand that there are very few work looking at childhood stunting and brain morphometry. However, it would have been beneficial to include a broader range of knowledge, such as studies examining malnutrition (which is closely related to stunting) and brain morphometry, or research on childhood stunting and its effects on other brain modalities, such as fMRI.

Answer: Thank you. We agree with the reviewer. We have referenced work done in preterm babies as this is the closest set available for comparison.

1.3. The study mentions addressing the research gap of limited data on the effects of early childhood stunting and catch-up growth on brain morphometry. However, the conclusion primarily emphasized the differences between the 'Never Stunted' and 'Always Stunted' children. The 'caught-up' groups were not discussed in detail, and there was limited interpretation of the findings related to these groups.

Answer: Our analysis revealed a decrease in the volumes of different brain regions across the groups, but we found statistically significant differences only between the 'Never Stunted' and 'Always Stunted' groups. However, we had explored the dose-response exhibited by brain regions due to catch-up growth and discussed the role of nutritional factors in catch-up growth. We acknowledge that in the "Discussion" section, we missed to include the catch-up growth groups in some comparisons, which has now been addressed. (474-476; 524-525)

2.1. Aside from age and gender, total intracranial volume is also an important covariate for volumetric analyses, which I don't think has been considered in the current study. In addition, I am curious whether the present study has considered the child's awake/asleep state during the MRI scan as a covariate.

Answer: Thank you. All children included in the study were asleep during MRI scanning. The current cohort is a low-and-middle-income cohort with a high percentage of microcephaly, mostly guided by parental head circumference – this is already published (Sindhu KN, Ramamurthy P, Ramanujam K, Henry A, Bondu JD, John SM, Koshy B, Bose A, Kang G and Mohan VR. Low head circumference during early childhood and its predictors in a semi-urban settlement of Vellore, Southern India. BMC Pediatrics2019.19:182. https://doi.org/10.1186/s12887-019-1553-0)

2.2. I understand that visual checks have been carried out visually, but I am wondering if post segmentation checks have been performed (e.g., using Qoala-T on the Freesurfer output) as the study has not mentioned and it will be important information.

Answer: Thank you for your suggestion. We have used Qoala-T on the freesurfer output for assessing the accuracy of visual quality checks of the scans. The process and the results have been described in the “MRI Image Acquisition and Processing” sub-section and in the “Results” section. (262-265)

3. Authors have stated that some restrictions will apply.

Answer: Thank you. We function within the Government of India regulations and are happy to share datasets in line with Indian Council of Medical Research guidelines.

4. There are some punctuation and grammatical errors, which may be a result of the long sentences. For example - "In a previous report from the same cohort, children who remained persistently stunted had a significant reduction in verbal, and total cognition scores by 4·6 points which corresponded to a 0·3 standard deviation difference compared to those never stunted, and is consistent with the previous literature [6, 17]." The comma is misplaced, and there are grammatical errors: "In a previous report from the same cohort, children who remained persistently stunted had significant reductions in verbal and total cognition scores, by 4·6 points which corresponded to a 0·3 standard deviation difference compared to those children who have never stunted, and are consistent with the previous literature [6, 17]."

Answer: The corrections have been made in the manuscript. (458-461)

5. I was curious on why the study looked till 9 years old, and not an older age since the study was interested in the catch-up growth on brain morphometry (for example, 12 years old is an age at which brain volume is suggested to continue growing till).

Answer: Thank you for your feedback. We are already being following up the participants till 12 years old and looking at their brain volumes, in lieu of their performances in psychological assessments. 

Reviewer #4: The work by Koshy et al to understand the morphometric changes associated with stunting and catch-up growth is interesting with a larger Indian cohort from a small town. Though I liked the approach but the study needs considerable improvement from the analysis point of view.

MAJOR

Abstract- Please specify the methodology used in the method section to let readers understand what was done before they understand the finding. Please specify multiple comparison etc here

Answer: We have added the methodology used in the study, under the “Methods” section in the “Abstract”. (152-154)

Introduction-

1. Please define stunting and the criteria to identify

2. The statement 1000days = 2 years is bit confusing.

Answer: We have added the explanation for stunting and have corrected the statement regarding 1000 days = 2 years in the “Introduction” section. (178-180)

Method

3. Every study should be an independent read. So please briefly mention the cohort details in main manuscript or in supplement.

Answer: We have added the cohort details in the main manuscript under the “Materials and Methods” section and the “Results” section. (226-232; 304-310)

4. Please mention the age at which the childrens were scanned. Additionally, provide details on How many children needed Triclofos- Please mention.

Answer: We have added the age details in the “MRI Image Acquisition and Processing” section and also the number of children needing Triclofos in the “Results” section. (319)

5. Are the Regional Brain volumes normalised. Most of regional level results will dissapear if Total brain volume is regressed.

Answer: Thank you. The current cohort is a low-and-middle-income cohort with a high percentage of microcephaly, mostly guided by parental head circumference – this is already published (Sindhu KN, Ramamurthy P, Ramanujam K, Henry A, Bondu JD, John SM, Koshy B, Bose A, Kang G and Mohan VR. Low head circumference during early childhood and its predictors in a semi-urban settlement of Vellore, Southern India. BMC Pediatrics2019.19:182. https://doi.org/10.1186/s12887-019-1553-0).

6. How was the Time of MRI aquisition adjusted.

Answer: We have not adjusted for Time of MRI acquisition in this study as the same protocol has been followed for all participants without any deviation. Similarly, there was no time differences reported after MRI acquisition for all participants. 

Result

Please provide all the details of ANNOVA analysis for each group. Please adjust the time of aquistion, TIV and results will get shifted from present. This will provide a robust picture of the analysis.

Answer: The ANOVA analyses were conducted separately to compare each stunting group with the healthy group. The p-value was determined for each group to demonstrate the decrease in brain volumes. Age and sex were adjusted as co-variates for each analysis. We have also explained these details in the "Materials and Methods" section, under the "Statistical Analyses" subsection. In the current study, we opted to focus on comparing individual stunting groups with the healthy group, subsequently only including the p-value rather than the DF, SS, and F-values. While including these would have provided a more comprehensive analysis had we assessed all groups collectively, for our specific investigation, individual group analyses were deemed more suitable for the looking into the impact of catch-up growth.

MINOR

(1) Writing needs to be improved through out the manuscript at several places- starting from abstract to discussion- For example- Amongst 251 children from the overall cohort, 178 children with a mean age of 9.54 were considered for further analysis.- It sounds like the volumetric parameter is assessed longitudinally. 

Please write the message in each Paragraphs clearly and directly. For example Paragraph 3 shows Neuroimaging is expensive which I agree, but authors themselves have used neuroimaging rather than any alternative.

Answer: We have made the recommended changes to the writing. (156-157; 207)

---

## [Decision Letter · Decision Letter 1]

14 Oct 2024

PONE-D-24-24149R1Childhood brain morphometry in children with persistent stunting and catch-up growthPLOS ONE

Dear Dr. Koshy,

Thank you for submitting your manuscript to PLOS ONE. After careful consideration, we feel that it has merit but does not fully meet PLOS ONE’s publication criteria as it currently stands. Therefore, we invite you to submit a revised version of the manuscript that addresses the points raised during the review process.

We look forward to receiving your revised manuscript.

Kind regards,

Cota Navin Gupta

Academic Editor

PLOS ONE

Journal Requirements:

Additional Editor Comments:

Thanks for addressing our queries..However in the "Response to Reviewers" doc the lines numbers where the changes have been made in the revised manuscript with reference to reviewer queries are haphazard and hard to track

Make sure that the line numbers in the "Response to Reviewers" doc match the changes made in the revised manuscript ?

Reviewers' comments:

Reviewer's Responses to Questions

**Comments to the Author**

1. If the authors have adequately addressed your comments raised in a previous round of review and you feel that this manuscript is now acceptable for publication, you may indicate that here to bypass the “Comments to the Author” section, enter your conflict of interest statement in the “Confidential to Editor” section, and submit your "Accept" recommendation.

Reviewer #2: All comments have been addressed

2. Is the manuscript technically sound, and do the data support the conclusions?

Reviewer #2: Partly

3. Has the statistical analysis been performed appropriately and rigorously? 

Reviewer #2: I Don't Know

4. Have the authors made all data underlying the findings in their manuscript fully available?

Reviewer #2: Yes

5. Is the manuscript presented in an intelligible fashion and written in standard English?

Reviewer #2: Yes

6. Review Comments to the Author

Reviewer #2: Most of the amendments suggested during the first round of review are addressed. The manuscript is technically sound and all the data presented are valuable information for carrying out further studies on child hood morphometry of

7. PLOS authors have the option to publish the peer review history of their article (what does this mean?). If published, this will include your full peer review and any attached files.

Reviewer #2: **Yes: **Apurba Patra

---

## [Author Response · Author response to Decision Letter 1]

18 Oct 2024

To

Dr. Cota Navin Gupta

Academic Editor

PLOS ONE

Dear Sir,

Thank you for your comments. Our responses are in CAPS below

THANK YOU. WE HAVE CHECKED THE REFERENCE LIST AGAIN

2. Thanks for addressing our queries..However in the "Response to Reviewers" doc the lines numbers where the changes have been made in the revised manuscript with reference to reviewer queries are haphazard and hard to track

Make sure that the line numbers in the "Response to Reviewers" doc match the changes made in the revised manuscript ?

THANK YOU. WE HAVE RECORRECTED BOTH DOCUMENTS

Let us know if any other corrections/changes are required.

Best

Beena

Dr. Beena Koshy

Professor

Developmental Paediatrics

CMC Vellore

---

## [Editor Report · Decision Letter 2]

13 Nov 2024

Childhood brain morphometry in children with persistent stunting and catch-up growth

PONE-D-24-24149R2

Dear Dr. Koshy,

We’re pleased to inform you that your manuscript has been judged scientifically suitable for publication and will be formally accepted for publication once it meets all outstanding technical requirements.

Kind regards,

Cota Navin Gupta

Academic Editor

PLOS ONE

www.iitg.ac.in/cngupta

Additional Editor Comments (optional):

Thanks for addressing the queries raised by me and reviewers...Congrats for this publication..Best wishes...Navin
---

## [Editor Report · Acceptance letter]

18 Nov 2024

PONE-D-24-24149R2 

PLOS ONE

Dear Dr. Koshy, 

I'm pleased to inform you that your manuscript has been deemed suitable for publication in PLOS ONE. Congratulations! Your manuscript is now being handed over to our production team.

Kind regards, 

on behalf of

Dr. Cota Navin Gupta 

Academic Editor

PLOS ONE